# Systemic steroid therapy for pneumonic chronic obstructive pulmonary disease exacerbation: A retrospective cohort study

**Akihiro Shiroshita**[1,2,3,4]*, **Keisuke Anan**[4,5], **Masafumi Takeshita**[3], **Yuki Kataoka**[4,6,7,8]

**1** Division of Epidemiology, Department of Medicine, Vanderbilt University School of Medicine, Nashville, Tennessee, United States of America, **2** Division of Allergy, Pulmonary and Critical Care Medicine, Department of Medicine, Vanderbilt University Medical Center, Nashville, Tennessee, United States of America, **3** Department of Respiratory Medicine, Ichinomiyanishi Hospital, Ichinomiya, Aichi, Japan, **4** Scientific Research Works Peer Support Group (SRWS-PSG), Osaka, Japan, **5** Division of Respiratory Medicine, Saiseikai Kumamoto Hospital, Kumamoto, Japan, **6** Department of Internal Medicine, Kyoto Min-Iren Asukai Hospital, Kyoto, Japan, **7** Section of Clinical Epidemiology, Department of Community Medicine, Kyoto University Graduate School of Medicine, Kyoto, Japan, **8** Department of Healthcare Epidemiology, Kyoto University Graduate School of Medicine/Public Health, Kyoto, Japan

* akihiro.shiroshita@vanderbilt.edu

**Data Availability Statement:** The data that support the findings of this study are not publicly available because the data was obtained from Real World Data, Co., Ltd.(RWD). but are available from the

## Abstract

The effectiveness of systemic steroid therapy on mortality in patients with pneumonic chronic obstructive pulmonary disease (COPD) exacerbation is unclear. We evaluated the association between systemic steroid therapy and 30-day mortality after adjusting for known confounders, using data from the Health, Clinic, and Education Information Evaluation Institute in Japan, which longitudinally followed up patients in the same hospital. We selected patients aged ≥40 years admitted for pneumonic COPD exacerbation. The exclusion criteria were censoring within 24 h, comorbidity with other respiratory diseases, and daily steroid use. Systemic steroid therapy was defined as oral/parenteral steroid therapy initiated within two days of admission. The primary outcome was the 30-day mortality rate. To account for known confounders, each patient was assigned an inverse probability of treatment weighting. The outcome was evaluated using logistic regression. Among 3,662 patients showing pneumonic COPD exacerbation, 30-day mortality in the steroid therapy and non-steroid therapy groups was 27.6% (169/612) and 21.9% (668/3,050), respectively. Systemic steroid therapy indicated a slightly higher estimated probability of 30-day mortality (difference in the estimated probabilities, 2.65%; 95% confidence interval, -1.23 to 6.54%, p-value = 0.181). Systemic steroid therapy within two days of admission was associated with higher 30-day mortality rates in pneumonic COPD exacerbation. Further validation studies based on chart reviews will be needed to cope with residual confounders.

## Introduction

Chronic obstructive pulmonary disease (COPD) and pneumonia are the leading causes of death worldwide [1]. The diagnostic criteria for COPD exacerbation and pneumonia overlap,

corresponding author, AS or the last author, YK with the permission of RWD (solution@rwdata.co.jp) on reasonable request.

**Funding:** The fee for the English editing service was covered by Real World Data, Co., Ltd (a profit organization). The publication fee was covered by a non-profit organization, the Japan Society for the Promotion of Science (Grants-in-Aid for Scientific Research [Kakenhi], grant number: 23K09582). The funder played no role in the study design, study execution, data analyses, data interpretation, or decision to submit the results.

**Competing interests:** AS received a research grant from Real World Data, Co., Ltd (a profit organization). and the Japan Society for the Promotion of Science, a non-profit organization (Grants-in-Aid for Scientific Research [Kakenhi] for this study. This does not alter our adherence to PLOS ONE policies on sharing data and materials.

and COPD exacerbation comorbid with pneumonia is called pneumonic COPD exacerbation [2]. The prognosis of pneumonic COPD exacerbation is worse than that of COPD exacerbation alone [3, 4]. However, a treatment strategy for pneumonic COPD exacerbations has not been established. In particular, the indications for systemic steroid therapy are a topic of long-standing debate. Previous randomized controlled trials (RCTs) targeted pneumonia alone or COPD exacerbation alone [5, 6]. Although many RCTs targeting COPD exacerbation excluded patients with pneumonia [5], many RCTs on pneumonia did not include enough patients with COPD, with unclear descriptions of the number of patients who also met the diagnostic criteria for COPD exacerbation [6].

Several retrospective cohort studies have evaluated the effectiveness of systemic steroid therapy for pneumonic COPD exacerbation [2, 7]. A single-center retrospective study by Scholl et al. suggested that the length of hospital stay may be shorter in the non-steroid users (6.0±4.0 days in the steroid group and 4.3±1.8 days in the non-steroid group) whereas there was no difference in 30-day mortality (3/67 [4%] in the steroid group and 0/38 [0%] in the non-steroid group) [7]. The multicenter retrospective study showed inconsistent results: a slightly decreased time to in-hospital death in the main analysis (hazard ratio [steroid/non-steroid group] 0.93, 95% confidence interval [CI], 0.92 to 0.94) and an opposite direction of the treatment effect in a sensitivity analysis (hazard ratio [steroid/non-steroid group] 1.1, 95% CI: 0.53 to 1.5). Thus, no study has provided definite evidence for the use of systemic steroid therapy. However, these studies had several risks of bias, such as unadjusted confounders, and had insufficient sample sizes for assessing mortality. Therefore, our study aimed to evaluate the association between systemic steroid therapy and mortality in patients with pneumonic COPD exacerbation in a more precise and valid manner. We used a large-scale database that contained a sufficient sample size to evaluate hard outcomes and used rigorous methodologies to account for confounders.

## Methods

### Study design

This retrospective cohort study used the Read World Data (RWD) database, a large-scale Japanese database maintained by the Health, Clinic, and Education Information Evaluation Institute (Kyoto, Japan) with support from Real World Data Co., Ltd. (Kyoto, Japan). This study was conducted in accordance with the Declaration of Helsinki [8]. The Institutional Review Board (IRB) of Ichinomiyanishi Hospital approved this study (approval number: 2022006), and the need for written informed consent was waived because the data in the RWD database had been de-identified. This article has been reported in accordance with the RECORD Statement (see S1 Table) [9].

### Patient selection

The RWD database includes cases from over 200 hospitals in Japan, ranging from primary to tertiary care hospitals. Its strength is that the patients were longitudinally followed up at the same hospital before and after hospitalization.

Our target population included patients aged ≥40 years who were admitted to a hospital because of pneumonic COPD exacerbation between October 2013 and June 2022. Our study period was between June 2022 and July 2023. Our patient selection algorithm was based on the 10th revision of the International Statistical Classification of Diseases and Related Health Problems (ICD-10). First, we extracted patients whose admission-precipitating diagnosis was bacterial pneumonia (ICD-10 codes J12, J13, J14, J15, J16, J18, J69, and P23) comorbid with COPD present at the time of admission (ICD-10 codes J44.1 and J44.9) and patients whose

admission-precipitating diagnosis was COPD exacerbation (ICD-10 code J44.1) comorbid with bacterial pneumonia present at the time of admission (ICD-10 codes J12, J13, J14, J15, J16, J18, J69, and P23). Second, we excluded patients aged ≤40 years, those who were censored (transferred or died) within 24 hours of admission, those with concomitant respiratory diseases on admission (asthma, heart failure [ICD-10 code J46], heart failure [ICD-10 code I50], pneumothorax [ICD-10 code J93], obstructive pneumonia [ICD-10 code J18], empyema [ICD-10 code J86]), and those with a history of daily systemic steroid use (systemic steroid use within 90 days before admission). Our patient selection algorithm had a sensitivity of 89% (95% CI: 71–98%) and specificity of 100% (95% CI: 88–100%) in a tertiary care hospital (2).

## Data extraction

We extracted patient demographic characteristics and diagnoses from the EF1 files submitted to the government for reimbursement of medical fees. These included age, admission date, sex, body mass index, exercise tolerability, dyspnea score, activities of daily living (ADL), presence of comorbidities, oxygen use on admission, mental status on admission, discharge date, and prognosis [10]. In addition, we extracted procedure and drug prescription data from claims and laboratory data from the data warehouse at each hospital. The drug prescription data were labeled by general name, World Health Organization Anatomical Therapeutic Chemical (ATC) Classification code, and the drug code assigned by the Ministry of Health Labour and Welfare to each drug [11]. The procedure data were labeled by the at-department procedure master for the medical service fee. The coding dictionary is summarized in the supplement (S2 Table).

## Exposure

The exposure was the use of oral/parenteral systemic steroid therapy on or one day after admission, regardless of the dose. The definition of systemic steroid therapy was based on ATC code H02 (corticosteroid for systemic use). Systemic steroid therapy was further divided into the following subcategories based on the drug codes proposed by the Ministry of Health, Labour and Welfare of Japan: triamcinolone, dexamethasone, hydrocortisone, prednisolone, and betamethasone.

## Outcome

The primary outcome was 30-day mortality. The follow-up period was calculated based on the date of admission (index date) and the date of the last follow-up visit at the same hospital. The patient's death was flagged, and the date of death was collected from the RWD database. If a patient was transferred to another hospital or died within 24 h, the data were censored. In addition, data for patients lost to follow-up within 30 days after the index date were also censored. The secondary outcomes were the time to in-hospital death and length of hospital stay. The time to in-hospital death was calculated using the index date (date of admission) and the date of in-hospital death. The length of hospital stay was calculated using the index date and the date of death until the end of hospitalization (the date of transfer to another hospital or discharge). These data were extracted from the EF files. In addition, we evaluated the number of cases showing acidosis (ICD-10 code: E872) and *Clostridioides difficile* colitis (ICD-10 code: A047) during hospitalization.

## Covariates

We selected known clinically important confounders: age (≥70 years), body mass index (<18.5, 18.5–25, >25 kg/m$^2$), a history of previous admission due to pneumonia or COPD

exacerbation within 90 days before the index date, ADL on admission (Barthel index: 0 to <20, 20 to <40, 40 to <85, ≥85), exercise tolerability and dyspnea (Hugh–Johns classification: 1 to 3, >3 to 6), mental status on admission (Japan Coma Scale score: 1–3, 10–30, 100–300), oxygen use on admission (procedure code: J024), blood eosinophil count (>300/μL), blood urea nitrogen (≥7 mmol/L or >19 mg/dL), and use of broad-spectrum antibiotics against *Pseudomonas aeruginosa* (S1 Fig) [12–14].

## Statistical analysis

Participants' characteristics were described by their systemic steroid status as frequencies and proportions for categorical variables and as means with standard deviations or medians and interquartile ranges (IQR) as appropriate for continuous variables. As for the primary outcome, we used a logistic regression model, and for the secondary outcomes, we used a Cox proportional hazard model for time to in-hospital death and a gamma regression model for length of hospital stay [15, 16]. We estimated the average differences in 30-day mortality and length of hospital stay for every single patient based on the presence of systemic steroid therapy. To account for the confounders, we calculated the propensity score, which is the probability of a patient receiving systemic steroid therapy conditioning on confounders, using a logistic regression model with adjustment for known confounders [17]. Each patient was assigned an average treatment-effect weight calculated using the inverse probability of treatment weighting (IPTW) method. We illustrated a love plot to confirm that the absolute standardized mean difference of each covariate was <0.1 in a weighted dataset and a histogram to confirm similar distributions of propensity scores in both treatment groups [18]. We then performed weighted outcome analyses according to the outcome type. Missing data were imputed using multiple imputations by chained equations, which created 100 imputed datasets and combined the estimate within each dataset [19]. Detailed information regarding our statistical analysis is provided in the Supporting information. Finally, we calculated the risk differences in the number of cases showing acidosis and *Clostridioides difficile* colitis.

For sensitivity analyses, we used the following statistical models for the primary outcome: 1) a different patient selection algorithm that did not exclude patients with other concomitant respiratory diseases or daily steroid users; 2) additionally excluding non-empirical antibiotic users; and 3) a complete case analysis. In Japanese administrative claims data, we can collect the A-DROP score, a modified version of CURB-65, at admission for patients with primary diagnosis of bacterial pneumonia [20]. Thus, we performed the main analysis among the subgroup of patients with the admission-precipitating diagnosis of bacterial pneumonia (ICD-10 codes J12, J13, J14, J15, J16, J18, J69, and P23) comorbid with COPD present at the time of admission (ICD-10 codes J44.1 and J44.9) and a sensitivity analysis additionally incorporating oxyhemoglobin saturation measured by pulse oximetry ≤90% or partial oxygen pressure in arterial blood ≤60 mmHg and systolic blood pressure ≤90 mmHg extracted from the A-DROP score. We used R version 4.2.3 (R Foundation, Vienna, Austria) for the statistical analyses. A two-sided p-value less than 0.05 was considered statistically significance.

## Results

### Descriptive analysis

In the RWD database, between October 2013 and June 2022, we identified 5,723 patients aged ≥40 years on admission who were categorized under the diagnostic codes for pneumonic COPD exacerbation (Fig 1). Among them, 2,061 patients were excluded based on the exclusion criteria, and 3,662 patients were included in our analysis. Table 1 summarizes the patient characteristics stratified by systemic steroid therapy. Systemic steroid therapy was administered to

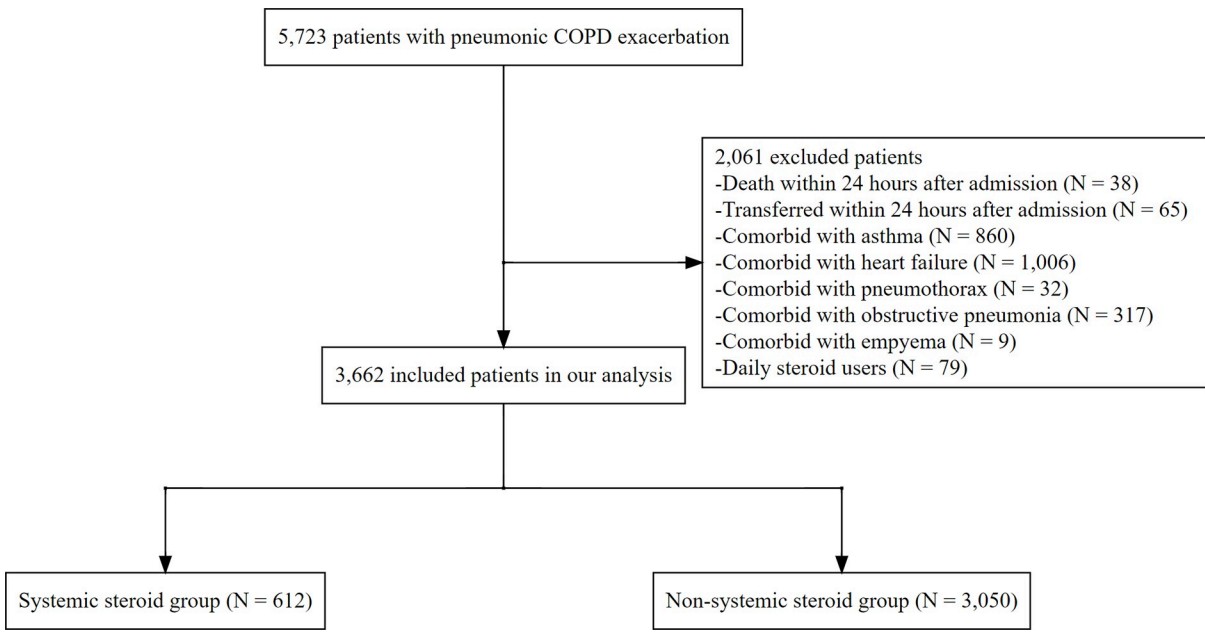

**Fig 1. Patient selection flow.** As for excluded patients, more than one option was chosen.

612 patients on or the day after admission, and methylprednisolone (53.9%) was the most frequently administered steroid drug (Table 2). Methylprednisolone was administered with a median dose of 200 mg (IQR: 120–280 mg) until hospital day 3 and a median dose of 280 mg (IQR: 153–480 mg) until hospital day 7. Prednisolone was administered with a median dose of 40 mg (IQR: 25–80 mg) until hospital day 3 and a median dose of 40 mg (IQR: 30–120 mg) until hospital day 7. Among the 3,050 patients in the non-systemic steroid therapy group, 859 (28.2%) were administered a systemic steroid within two days after admission. Almost all patients (94.3%) were administered empirical antibiotics, and 24.6% (21.0% in non-systemic steroid users and 33.7% in systemic steroid users) were administered broad-spectrum antibiotics within two days after admission. Among the patients who were not empirically administered broad-spectrum antibiotics, 25.4% (26.2% in non-systemic steroid users and 20.7% in systemic steroid users) were additionally administered broad-spectrum antibiotics until hospital day 7.

## Primary and secondary outcomes

None of the patients were lost to follow-up within 30 days after the day of admission. The 30-day mortality rate was 668/3,050 (21.9%) in the non-steroid therapy group and 169/612 (27.6%) in the steroid therapy group. In both the imputed models and complete case analyses, the covariates were well-balanced after propensity-score weighting (Figs 2 and 3). The analysis results for 30-day mortality are summarized in Table 3. Our primary analysis (multiple imputation + propensity-score weighting + logistic regression) showed that the difference (steroid therapy group vs. non-steroid therapy group) in the probabilities of death at 30 days was 2.65% (95% CI: -1.23% to 6.54%, p-value = 0.181). These results are similar to those of the sensitivity analysis (Table 3). As for the time to in-hospital death, the estimated hazard ratio (steroid therapy group vs. non-steroid therapy group) was 1.30 (95% CI: 1.00 to 1.68, p-value = 0.049). The difference in the length of hospital stay was -1.87 days (95% CI: -3.43 to -0.31 days, p-value = 0.019).

**Table 1. Patient characteristics.**

| | Non-systemic steroid users | Systemic steroid users | Overall |
|---|---|---|---|
| | (N* = 3,050) | (N = 612) | (N = 3,662) |
| Age (mean [SD]) | 79.1 (7.4) | 79.0 (7.2) | 79.1 (7.4) |
| Male (%) | 2,703 (88.6) | 550 (89.9) | 3,253 (88.8) |
| Body mass index (mean (SD†)) | 18.0 (7.0) | 16.4 (8.6) | 17.7 (7.3) |
| The activity of daily living on admission (%) | | | |
| Fully dependent | 1,736 (56.9) | 315 (51.5) | 2,051 (56.0) |
| Partially dependent | 1,182 (38.8) | 238 (38.9) | 1,420 (38.8) |
| Partially independent | 97 (3.2) | 48 (7.8) | 145 (4.0) |
| Independent | 35 (1.1) | 11 (1.8) | 46 (1.3) |
| Long-term oxygen therapy before admission (%) | | | |
| Home oxygen therapy | 34 (1.1) | 6 (1.0) | 40 (1.1) |
| Home mechanical ventilation | 2 (0.1) | 0 (0) | 2 (0.1) |
| Mental status on admission (%) | | | |
| Normal | 2,540 (83.3) | 482 (78.8) | 3,022 (82.5) |
| 1-digits | 423 (13.9) | 92 (15.0) | 515 (14.1) |
| 2-digits | 58 (1.9) | 21 (3.4) | 79 (2.2) |
| 3-digits | 29 (1.0) | 17 (2.8) | 46 (1.3) |
| Hugh–Johns classification > 3 (%) | 1,598 (52.4) | 404 (66.0) | 2,002 (54.7) |
| Charlson Comorbidity Index (median [IQR‡]) | 1.0 [1.0, 3.0] | 1.0 [1.0, 2.0] | 1.0 [1.0, 3.0] |
| Cor pulmonale (%) | 16 (0.5) | 4 (0.7) | 20 (0.5) |
| Bronchiectasis (%) | 38 (1.2) | 3 (0.5) | 41 (1.1) |
| Diabetes (%) | 498 (16.3) | 107 (17.5) | 605 (16.5) |
| Lung cancer (%) | 249 (8.2) | 37 (6.0) | 286 (7.8) |
| Previous hospitalization due to pneumonic or COPD exacerbation within 90 days (%) | 321 (10.5) | 51 (8.3) | 372 (10.2) |
| Daily antacid users (%) | 1,956 (64.1) | 448 (73.2) | 2,404 (65.6) |
| Blood eosinophil count (median [IQR]) | 42.8 [7.0, 150.9] | 19.1 [0.0, 85.2] | 37.5 [0.0, 140.2] |
| Missing data (%) | 480 (15.7) | 74 (12.1) | 475 (13.0) |
| Blood urea nitrogen (median [IQR]) | 17.9 [13.5, 23.5] | 18.5 [14.0, 25.2] | 18.0 [13.7, 24.0] |
| Missing data (%) | 162 (5.3) | 19 (3.1) | 181 (4.9) |
| Serum albumin (median [IQR]) | 3.3 [2.8, 3.6] | 3.3 [2.9, 3.7] | 3.3 [2.9, 3.6] |
| Oxygen use on admission (%) | 1,730 (56.7) | 391 (63.9) | 2,121 (57.9) |
| Vasopressor use on admission (%) | 24 (0.8) | 32 (5.2) | 56 (1.5) |
| Tracheal intubation (%) | 48 (1.6) | 31 (5.1) | 79 (2.2) |
| Mechanical ventilation§ (%) | 184 (6.0) | 86 (14.1) | 270 (7.4) |
| Length of hospital stay (median [IQR]) | 13.0 [8.0, 22.0] | 13.0 [8.0, 21.0] | 13.0 [8.0, 22.0] |
| In-hospital mortality (%) | 324 (10.6) | 84 (13.7) | 408 (11.1) |
| 30-day mortality (%) | 668 (21.9) | 169 (27.6) | 837 (22.9) |

*: N = number

†: SD = standard deviation

‡: IQR = interquartile range; §: mechanical ventilation included invasive/non-invasive ventilation, high-flow nasal cannula, and continuous positive airway pressure

The risk difference (steroid group vs. non-steroid group) in the number of acidosis cases was -0.12% (0/612 vs. 6/3050, 95% CI: -0.35% to -0.04%), and that for *Clostridioides difficile* colitis was -0.04% (5/612 vs. 26/3050, 95% CI: -0.82% to 0.75%). We could not evaluate subcategories of acidosis, such as diabetic ketoacidosis and lactic acidosis.

**Table 2. Subcategories of systemic steroid therapy.**

|  | Number of patients (%) | Median dose and IQR* (mg) until hospital day 3 | Median dose and IQR (mg) until hospital day 3 |
|---|---|---|---|
| Betamethasone | 57 (9.3) | 12 (5–20) | 12 (4–28) |
| Dexamethasone | 24 (3.9) | 4 (2–6) | 5 (2–10) |
| Methylprednisolone | 330 (53.9) | 200 (120–280) | 280 mg (153–480) |
| Prednisolone | 139 (22.7) | 40 (25–80) | 40 (30–120) |
| Hydrocortisone | 27 (4.4) | 400 (300–600) | 600 (300–750) |

*:IQR = interquartile range

Because we selected systemic steroid therapy on the day or the next day of admission, more than one option was chosen.

## Discussion

This was the first study validly evaluating the effectiveness of systemic steroid therapy on mortality in patients with pneumonic COPD exacerbation. In our study, the use of systemic steroid therapy was associated with a slightly higher probability of death at 30 days and a higher hazard ratio for in-hospital death. Although these results were associated with a shorter length of

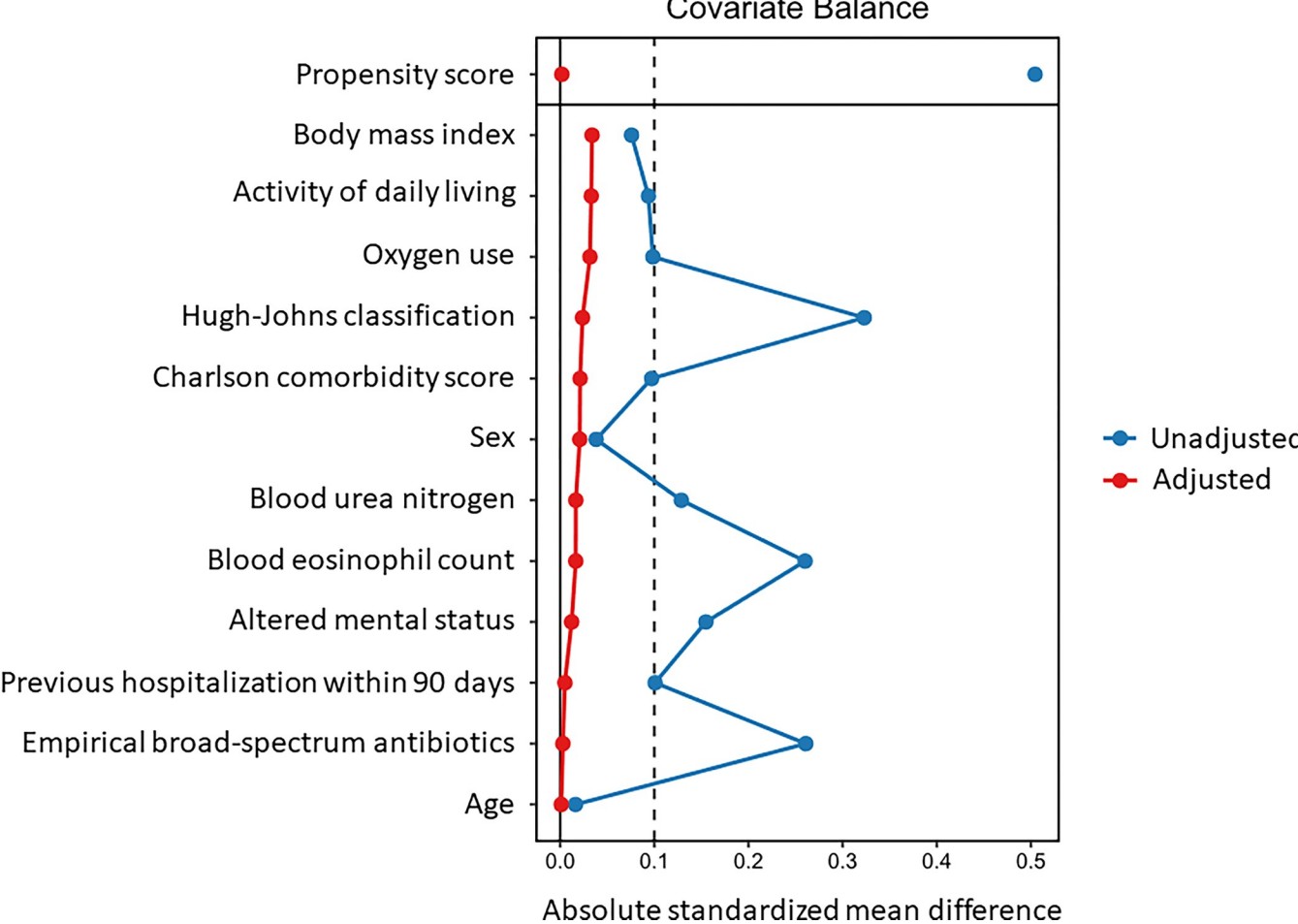

**Fig 2. Love plot in the complete case analysis.** The covariates were well-balanced (absolute standardized mean difference < 0.1) after the average treatment-effect weight was assigned to each patient. We confirmed the balance in both imputed models and complete case analyses using the same figures. As for the imputed models, the balance was checked within each imputed dataset.

## Distributional balance for propensity score

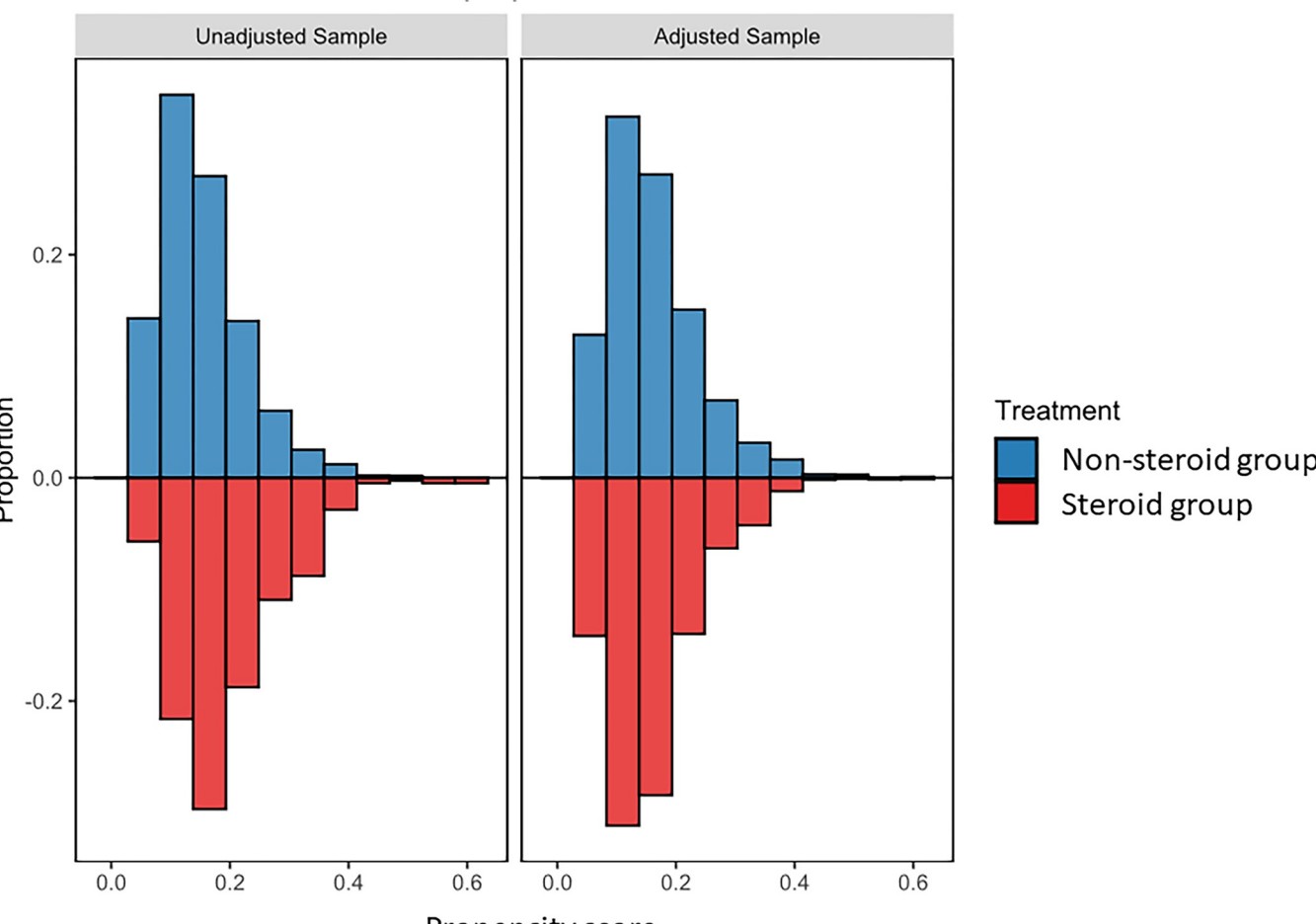

**Fig 3. Histogram of propensity scores in the complete case analysis.** This histogram shows that the distributions of propensity scores in the non-systemic steroid and systemic steroid groups were well-overlapped. We confirmed the distribution of propensity scores in both imputed models and complete case analyses using the same figures. As for the imputed models, the distribution was checked within each imputed dataset.

hospital stay, they may have been influenced by the higher mortality rate. The incidence of adverse reactions, including acidosis and *Clostridioides difficile* colitis, was not substantially different between the systemic and non-systemic steroid groups.

Our results suggest the futility of systemic steroid therapy for pneumonic COPD exacerbation. The inflammation profiles underlying pneumonic COPD exacerbation remain unclear, but they have not been focused on in previous RCTs evaluating pneumonic alone or COPD exacerbation alone [21]. Previous single-center and multicenter retrospective cohort studies also did not reveal any benefits of systemic steroid therapy for pneumonic COPD exacerbation [2, 7]. However, their major limitation was the lack of a sufficient sample size to evaluate mortality. The current study used a nationwide Japanese database to overcome this limitation. We used a validated patient selection algorithm and accounted for many confounding factors. The use of systemic steroid therapy was associated with a slightly higher 30-day mortality rate and a higher hazard ratio of time to in-hospital death. Our sensitivity analysis showed similar results. We conclude that there is no evidence supporting the use of systemic steroid therapy for pneumonic COPD exacerbations.

**Table 3. Statistical analysis results for the 30-day mortality.**

| | Difference in the predicted probabilities of death at 30 days (Steroid therapy group vs. non-steroid therapy group [%], 95% confidence interval, p-value) |
|---|---|
| **Primary analysis** | |
| Multiple imputation + propensity-score weighting + logistic regression (N* = 3,662) | 2.65% (95% CI: -1.23 to 6.54%, p-value = 0.181) |
| **Sensitivity analyses** | |
| Complete case analysis (propensity-score weighting + logistic regression, N = 2,648) | 3.52% (95% CI: 1.50 to 5.53%, p-value < 0.001) |
| Including comorbidity with other respiratory diseases (multiple imputation + propensity-score weighting + logistic regression, N = 5,679) | 1.95% (95% CI: -0.95 to 4.85%, p-value = 0.188) |
| Including patients who were administered empirical antibiotics (multiple imputation + propensity-score weighting + logistic regression, N = 3,662) | 3.98% (95% CI: -0.01 to 7.96%, p-value = 0.050) |
| Including only patients with an admission-precipitating diagnosis of bacterial pneumonia comorbid with COPD present at the time of admission and incorporating the A-DROP score (multiple imputation + propensity-score weighting + logistic regression, N = 2,669) | 0.09% (95% CI: -4.37 to 4.18%, p-value = 0.966) |

*: N = number

Regarding the time to in-hospital death, the direction of the treatment effect of systemic steroid therapy was inconsistent with our previous study (the current study, HR 1.30; the previous study, HR 0.93) [2], which may reflect confounding by indication. For example, our previous study did not control for the baseline function of COPD as a confounder. In addition, antibiotic therapy was not assessed in the previous study. These unadjusted potential confounding factors may have skewed the direction of the treatment effect away from the null. In the current study, we overcame these limitations and proposed a robust result.

Regarding adverse events, although the sample size of our study was inadequate and there was no substantial difference in the number of cases showing acidosis and *Clostridioides difficile* colitis, short-term systemic steroid therapy is associated with various adverse outcomes such as sepsis, venous thromboembolism, and fracture [22, 23]. We cannot elucidate the mechanisms by which systemic steroid therapy is associated with worse outcomes; however, one possible hypothesis is that these outcomes are adverse reactions to systemic steroid therapy. On the basis of our study results, the mortality rate may improve by avoiding the unnecessary use of systemic steroid therapy.

Our results may also reflect treatment changes during hospitalization. During hospitalization, 859/3,050 patients (28.2%) in the non-steroid therapy group were administered systemic steroid therapy two days after admission. In addition, among the patients who were not empirically administered broad-spectrum antibiotics, 26.2% of non-systemic steroid users and 20.7% of systemic steroid users were additionally administered broad-spectrum antibiotics two days after admission. We cannot determine whether physicians decided to change the treatment due to an exacerbation of the infection or a subsequent occurrence of a new infection. In addition, there could be time-varying confounders between the treatment change and the outcome. Thus, we could not assess treatment failure as an outcome. Nevertheless, at this point, our results do not support the routine use of systemic steroid therapy on the day of or the day after admission. Evaluation of the effect of systemic steroid therapy during the entire

hospital stay requires adjustments for time-varying confounders as well as a much larger sample size.

Our study had several limitations. First, the diagnosis of pneumonic COPD exacerbation was based on the ICD-10 algorithm, which has been validated only in a tertiary care hospital [2]. In addition, the gold standard for COPD diagnosis in the validation study was the physician's description in the medical record. It was unclear whether spirometry or pulmonary function tests were performed to diagnose COPD. The sensitivity and specificity of our algorithm for detecting pneumonic COPD may differ depending on the outcome status. For example, the admission-precipitating diagnosis may have changed from pneumonia or COPD exacerbation due to subsequent severe disease during hospitalization. This differential selection bias may have skewed the true effects of systemic steroid therapy to the null. Thus, standardization of the diagnostic protocol is needed in future studies. Second, the adjustment for confounders may have been insufficient. We adjusted for known measured confounders and proxies for unmeasured confounders, such as baseline factors (e.g., Hugh–Johns classification and ADL) and severity on admission (e.g., use of vasopressors and oxygen). In addition, we confirmed the same direction of treatment effects among the patients with pneumonia comorbid with COPD and/or COPD exacerbation who had the A-DROP score. However, other unmeasured confounders may still exist. For example, although the number of patients comorbid with Cor pulmonale was quite small, the accuracy of the disease code was questionable. The other examples were pneumococcal and influenza vaccines as well as pulmonary function test results and image findings. When they were positively associated with the use of systemic steroid therapy and with higher mortality, our results may have been biased toward harmful. RCTs are warranted to address unmeasured confounders. Finally, in the RWD data, a patient was followed only in the same hospital, and we could not link the information in different hospitals. If patients received home oxygen therapy and home mechanical ventilation from another hospital, we could not detect them. It could be related to residual confounding factors.

Although a large-scale retrospective study and a randomized controlled study can have additional value, a larger-scale prospective cohort study would be desirable. First, it can avoid misclassification of pneumonic COPD exacerbation. Many physicians do not routinely perform chest computed tomography (CT) scans. In our dataset, pneumonia was detected by chest X-ray in some patients and by chest CT scan in others. The heterogeneity of patients with pneumonic COPD exacerbation would undermine internal and external validity. Researchers should establish a standardized protocol to detect pneumonic COPD exacerbation. In addition, some variables can be collected with difficulty in retrospective studies. For example, no study could use baseline severity of COPD such as airflow limitation examined by the pulmonary function test and exertional dyspnea examined by the modified Medical Research Council Dyspnea Scale and COPD Assessment Test, because of the large number of missing data. Although a randomized controlled study can overcome the confounding issue by indication, it would require high human and financial costs considering the low effect size of systemic steroid therapy. For these reasons, a large-scale prospective study with a predefined protocol would be the most valuable option.

## Conclusions

Systemic steroid therapy on or the day after admission may not improve 30-day mortality or in-hospital mortality in patients with exacerbation of pneumonic COPD. Moreover, physicians should not routinely use systemic steroid therapy for cases showing pneumonic COPD exacerbation. Further validation studies based on clinical records will overcome our study limitation of residual confounders.

## Supporting information

**S1 Fig. Direct acyclic graph.** In our statistical analysis models, the following confounders were adjusted: age ($\geq$70 years old), body mass index (<18.5, 18.5–25, >25), presence of the previous admission due to pneumonia or COPD exacerbation within 90 days before the index date, the activity of daily living on admission (Barthel index: 0 to <20, 20 to <40, 40 to <85, $\geq$85), exercise tolerability and dyspnea (Hugh-Johns classification: 1 to 3, 3< to 6), mental status on admission (Japan Coma Scale: 1–3, 10–30, 100–300), oxygen use on admission (procedure code: J024), blood eosinophil count (>300/µL), and blood urea nitrogen ($\geq$7 mmol/L), and use of broad-spectrum antibiotics.
(PNG)

**S1 Table. Empirical antibiotics used in the included observations.** Because some patients received combination of two or more antibiotics, the sum of each column is over 100%.
(DOCX)

**S2 Table. The RECORD statement.**
(DOCX)

**S3 Table. Coding dictionary.**
(DOCX)

## Acknowledgments

The authors thank the Health, Clinic, and Education Information Evaluation Institute for the database development for the study.

## Author Contributions

**Conceptualization:** Akihiro Shiroshita, Keisuke Anan, Masafumi Takeshita, Yuki Kataoka.

**Data curation:** Akihiro Shiroshita.

**Formal analysis:** Akihiro Shiroshita.

**Funding acquisition:** Akihiro Shiroshita.

**Investigation:** Akihiro Shiroshita, Keisuke Anan, Yuki Kataoka.

**Methodology:** Akihiro Shiroshita, Keisuke Anan, Yuki Kataoka.

**Project administration:** Akihiro Shiroshita, Masafumi Takeshita, Yuki Kataoka.

**Resources:** Akihiro Shiroshita, Yuki Kataoka.

**Software:** Akihiro Shiroshita.

**Supervision:** Keisuke Anan, Masafumi Takeshita, Yuki Kataoka.

**Validation:** Keisuke Anan, Masafumi Takeshita, Yuki Kataoka.

**Visualization:** Akihiro Shiroshita, Keisuke Anan, Yuki Kataoka.

**Writing – original draft:** Akihiro Shiroshita, Keisuke Anan, Yuki Kataoka.

**Writing – review & editing:** Akihiro Shiroshita, Keisuke Anan, Masafumi Takeshita, Yuki Kataoka.

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
