## [Decision Letter · Decision Letter 0]

29 Jun 2023

PONE-D-23-10593Systemic steroid therapy for pneumonic chronic obstructive pulmonary disease exacerbation: A retrospective cohort studyPLOS ONE

Dear Dr. Shiroshita,

Thank you for submitting your manuscript to PLOS ONE. After careful consideration, we feel that it has merit but does not fully meet PLOS ONE’s publication criteria as it currently stands. Therefore, we invite you to submit a revised version of the manuscript that addresses the points raised during the review process.

We look forward to receiving your revised manuscript.

Kind regards,

Dong Keon Yon, MD, FACAAI, FAAAAI

Academic Editor

PLOS ONE

Journal Requirements:

"I have read the journal's policy and the authors of this manuscript have the following competing interests: AS received a research grant from Real World Data, Co., Ltd for this study. The other authors had nothing to declare."

We note that you received funding from a commercial source: Real World Data, Co., Ltd.

Within this Competing Interests Statement, please confirm that this does not alter your adherence to all PLOS ONE policies on sharing data and materials by including the following statement: ""This does not alter our adherence to PLOS ONE policies on sharing data and materials.” (as detailed online in our guide for authors http://journals.plos.org/plosone/s/competing-interests).  If there are restrictions on sharing of data and/or materials, please state these. Please note that we cannot proceed with consideration of your article until this information has been declared. 

Additional Editor Comments:

Thank you for submitting your manuscript. The reviewers and I believe it is of potential value for our readers. However, the reviewers have raised a number of very important issues, and their excellent comments will need to be adequately addressed in a revision before the acceptability of your manuscript for publication in the Journal can be determined. We cannot guarantee that your revised paper will be chosen for publication; this would be solely based on how satisfactorily you have addressed the reviewer comments.

#1. Based on reviewers comment, please cite these statistical guideline papers.

- A) https://doi.org/10.1093/ckj/sfab158

- B) DOI: https://doi.org/10.54724/lc.2023.e8

#2. Please add the P value statement.

A two-sided P value less than 0.05 considered statistical significance.

#3. R Core Team -> R Foundation

#4. This is an excellent paper.

Reviewers' comments:

Reviewer's Responses to Questions

**Comments to the Author**

1. Is the manuscript technically sound, and do the data support the conclusions?

Reviewer #1: Yes

Reviewer #2: Yes

2. Has the statistical analysis been performed appropriately and rigorously? 

Reviewer #1: Yes

Reviewer #2: Yes

3. Have the authors made all data underlying the findings in their manuscript fully available?

Reviewer #1: Yes

Reviewer #2: Yes

4. Is the manuscript presented in an intelligible fashion and written in standard English?

Reviewer #1: Yes

Reviewer #2: Yes

5. Review Comments to the Author

Reviewer #1: The study is no doubt good but , I have a few queries which I would wish the authors explain:

1) How was the diagnosis of COPD made in these patients? Was it clinical or spirometry based?

2) How many of these patients had PAH on Echo as that could also affect the outcome in terms of mortality?

3) How many of these patients were on LTOT on BiPAP at home prior to admission because this group of patients is assumed belong to more severe category ?

4) Was pneumococcal or influenza vaccination history taken into account?

5) The severity of the COPD patients based on FEV1 is also not mentioned in the study which could have affected the results.

6) If a diagnosis of COPD was made on PFT, how many of these patients had reversibility?

7) How many patients had multilobar involvement on x ray and was any bacteriological correlation made?

8) Whst was the choice of antibiotics in these patients?

Thank you.

Reviewer #2: Dear Authors,

I have now concluded the review of your manuscript entitled "Systemic steroid therapy for pneumonic chronic obstructive pulmonary disease exacerbation: A retrospective cohort study."

This study, which investigates the association between systemic steroid therapy and 30-day mortality among patients with pneumonic COPD exacerbations, is undeniably fascinating. The manuscript is generally well-written and provides valuable insights.

However, to enhance the overall quality and impact of your manuscript, I recommend considering the following suggestions:

1. In the Introduction section, you have referred to several relevant studies. It would be helpful if you could elaborate on the results of these studies, incorporating their effect sizes to provide a comprehensive understanding of the current knowledge in the field.

2. There is a possibility that patients receiving systemic steroids were more severely ill, thereby predisposing them to higher mortality irrespective of steroid therapy. Despite efforts to adjust for known confounders, residual confounding factors might persist. I propose undertaking a sensitivity analysis to account for the severity of illness.

3. Pneumonic COPD exacerbations are typically attributed to infections. The immunosuppressive effect of systemic steroids could potentially exacerbate these infections. Could you clarify how the potential for increased infection was controlled in your study?

4. In the ‘Statistical Analysis’ section, the paragraph is quite long but there are no references for selecting your method. Please justify yourself about selecting those methods, or refer to some articles on how to select proper regression analysis for continuous independent variables in medical research. Also, reference about propensity score for causal inference and reducing the confounding effects are recommended.

5. Lastly, I would appreciate it if you could highlight the future scope of your study. Your comprehensive limitation discussion indicates several avenues for future research, and these potential directions should be elucidated.

I hope you find these suggestions beneficial in strengthening your manuscript. I am looking forward to your revised submission.

6. PLOS authors have the option to publish the peer review history of their article (what does this mean?). If published, this will include your full peer review and any attached files.

Reviewer #1: No

Reviewer #2: No

---

## [Author Response · Author response to Decision Letter 0]

3 Aug 2023

Point-by-Point Responses to the Comments of the Reviewers

Additional Editor Comments:

Thank you for submitting your manuscript. The reviewers and I believe it is of potential value for our readers. However, the reviewers have raised a number of very important issues, and their excellent comments will need to be adequately addressed in a revision before the acceptability of your manuscript for publication in the Journal can be determined. We cannot guarantee that your revised paper will be chosen for publication; this would be solely based on how satisfactorily you have addressed the reviewer comments.

Response: We thank the editor for the constructive comments and suggestions that helped improve our manuscript. We have revised the manuscript accordingly.

#1. Based on reviewers comment, please cite these statistical guideline papers.

- A) https://doi.org/10.1093/ckj/sfab158

- B) DOI: https://doi.org/10.54724/lc.2023.e8

Response: We have inserted these two important articles in the manuscript.

References [16] and [17]

#2. Please add the P value statement.

A two-sided P value less than 0.05 considered statistical significance.

Response: We have added the description of the p-value in the methods section and the p-values obtained by our statistical analyses in addition to the 95% confidence intervals.

Methods

Page 9, Lines 191–192: “A two-sided p-value less than 0.05 was considered statistically significant.” 

Results

Page 13, Lines 229–236: “Our primary analysis (multiple imputation + propensity-score weighting + logistic regression) showed that the difference (steroid therapy group vs. non-steroid therapy group) in the probabilities of death at 30 days was 2.65% (95% CI: -1.23% to 6.54%, p-value = 0.181). These results are similar to those of the sensitivity analysis (Table 3). As for the time to in-hospital death, the estimated hazard ratio (steroid therapy group vs. non-steroid therapy group) was 1.30 (95% CI: 1.00 to 1.68, p-value = 0.049). The difference in the length of hospital stay was -1.87 days (95% CI: -3.43 to -0.31 days, p-value = 0.019).”

Table 3. Statistical analysis results for the 30-day mortality

 Difference in the predicted probabilities of death at 30 days (Steroid therapy group vs. non-steroid therapy group [%], 95% confidence interval, p-value)

Primary analysis

Multiple imputation + propensity-score weighting + logistic regression (N* = 3,662) 2.65% (95% CI: -1.23 to 6.54%, p-value = 0.181)

Sensitivity analyses

Complete case analysis (propensity-score weighting + logistic regression, N = 2,648) 3.52% (95% CI: 1.50 to 5.53%, p-value < 0.001)

Including comorbidity with other respiratory diseases (multiple imputation + propensity-score weighting + logistic regression, N = 5,679) 1.95% (95% CI: -0.95 to 4.85%, p-value = 0.188)

Including patients who were administered empirical antibiotics (multiple imputation + propensity-score weighting + logistic regression, N = 3,662) 3.98% (95% CI: -0.01 to 7.96%, p-value = 0.050)

Including only patients with an admission-precipitating diagnosis of bacterial pneumonia comorbid with COPD present at the time of admission and incorporating the ADROP score (multiple imputation + propensity-score weighting + logistic regression, N = 2,669) 0.09% (95% CI: -4.37 to 4.18%, p-value = 0.966)

#3. R Core Team -> R Foundation

Response: We have modified the description of the statistical software, R.

#4. This is an excellent paper.

Response: Thank you so much again for your compliment.

Other corrections:

1. We inattentively forgot to include blood urea nitrogen as a covariate in some statistical models. Thus, we have re-run all the analyses and updated the results. Because we used a different computer, the analysis results were similar but a little different. The direction of the estimates was the same, and we did not change our conclusion, but again, we apologize for our inattentive care. 

2. AS has updated his funding information. The fee for the English editing service was covered by Real World Data, Co., Ltd (a profit organization). The publication fee will be covered by a non-profit organization, the Japan Society for the Promotion of Science (Grants-in-Aid for Scientific Research [Kakenhi], grant number: 23K09582). The other authors have nothing to disclose. This change was written in the cover letter, and the funding and conflicts of interest sections of the main text.

3. We have additionally described the data availability statement in detail. It was reflected in the cover letter and the data availability statement in the main text.

Reviewer 1:

The study is no doubt good but , I have a few queries which I would wish the authors explain.

Response: We thank the reviewer for reviewing our manuscript and providing valuable comments. We have revised our manuscript accordingly.

1) How was the diagnosis of COPD made in these patients? Was it clinical or spirometry based?

Because our dataset was based on administrative claims data, we were not able to determine how the physicians diagnosed patients with COPD. To mitigate the selection bias, we performed a validation study evaluating the accuracy of the disease name used for claims data by referring to the physicians’ clinical diagnosis in the electrical medical records. It showed that our patient selection algorithm could accurately detect patients with pneumonic COPD exacerbation with a sensitivity of 89% (95% CI: 71–98%) and a specificity of 100% (95% CI: 88–100%). However, it is still unclear whether spirometry or the pulmonary function test was performed to diagnose COPD. We have added this to the limitations part of the discussion section.

Reference:

Shiroshita A, Shiba H, Tanaka Y, Nishi A, Sato K, Shirakawa C, et al. Effectiveness of steroid therapy on pneumonic chronic obstructive pulmonary disease exacerbation: a multicenter, retrospective cohort study. Int J Chron Obstruct Pulmon Dis. 2020;15: 2539-2547

Discussion

Page 17, Lines 305–307: “In addition, the gold standard for COPD diagnosis in the validation study was the physician’s description in the medical record. It was unclear whether spirometry or pulmonary function tests were performed to diagnose COPD.” 

2) How many of these patients had PAH on Echo as that could also affect the outcome in terms of mortality?

4) Was pneumococcal or influenza vaccination history taken into account?

Response: We are very sorry to inform you that we were not able to collect data on echocardiography or vaccine status because of the nature of administrative claims data. Although our patient selection algorithm for pneumonic COPD exacerbation excluded heart failure, we have additionally reviewed the presence of the disease name “Cor pulmonale” (ICD-10: I27) as a comorbidity at hospital admission for pneumonic COPD exacerbation. The number of patients comorbid with Cor pulmonale was 20/3,662 (0.5%) after excluding heart failure. We have added the information in Table 1. The number was quite small and would not have influenced our study results. However, because the accuracy of the disease name “Cor pulmonale” was unclear, we may have underestimated its prevalence. Thus, we have added it to the discussion section as a limitation. In addition, we have described the vaccine status as a potential confounder in the discussion.

Table 1. Patient characteristics

 Non-systemic steroid users

(N = 3,050) Systemic steroid users

(N = 612) Overall

(N = 3,662)

Cor pulmonale (%) 16 (0.5) 4 (0.7) 20 (0.5)

Discussion

Page 18, Lines 318–323: “However, other unmeasured confounders may still exist. For example, although the number of patients comorbid with Cor pulmonale was quite small, the accuracy of the disease code was questionable. The other examples were pneumococcal and influenza vaccines, as well as pulmonary function test results and image findings. When they were positively associated with the use of systemic steroid therapy and with higher mortality, our results may have been biased toward harmful.” 

3) How many of these patients were on LTOT on BiPAP at home prior to admission because this group of patients is assumed belong to more severe category ?

Response: We have added the number of patients who received home oxygen therapy and home mechanical ventilation to Table 1. As with Cor pulmonale, the number of patients was quite small and would not have influenced our results. However, in our dataset, a patient was followed only in the same hospital, and we could not link the information in different hospitals. Thus, information on home oxygen therapy and home mechanical ventilation could be missing. We have added it to the discussion section.

Table 1. Patient characteristics

 Non-systemic steroid users

(N = 3,050) Systemic steroid users

(N = 612) Overall

(N = 3,662)

Long-term oxygen therapy before admission 

 Home oxygen therapy 34 (1.1) 6 (1.0) 40 (1.1)

 Home mechanical ventilation 2 (0.1) 0 (0) 2 (0.1)

Discussion

Page 18, Lines 323–327: “Finally, in the RWD data, a patient was followed only in the same hospital, and we could not link the information in different hospitals. If patients received home oxygen therapy and home mechanical ventilation from another hospital, we could not detect them. It could be related to residual confounding factors.”

5) The severity of the COPD patients based on FEV1 is also not mentioned in the study which could have affected the results.

6) If a diagnosis of COPD was made on PFT, how many of these patients had reversibility?

7) How many patients had multilobar involvement on x ray and was any bacteriological correlation made?

Response: As per your comment #2 as well, we were not able to obtain these variables because our data set was based on administrative claims data. It does not contain the results of pulmonary function tests or imaging tests. We have added them as limitations to the discussion section.

Discussion

Page 18, Lines 318–323: “However, other unmeasured confounders may still exist. For example, although the number of patients comorbid with Cor pulmonale was quite small, the accuracy of the disease code was questionable. The other examples were pneumococcal and influenza vaccines as well as pulmonary function test results and image findings. When they were positively associated with the use of systemic steroid therapy and with higher mortality, our results may have been biased toward the harmful.” 

8) Whst was the choice of antibiotics in these patients?

Response: We have summarized the empirical antibiotics used in the included patients as a supplementary table based on the WHO Anatomical Therapeutic Chemical code.

S3 Table. Empirical antibiotics used in the included patients

 Empirical antibiotics among non-systemic steroid users

(N* = 2,845) Empirical antibiotics among systemic steroid users

(N = 607) Overall

(N = 3,452)

Aminoglycosides 22 (0.8%) 7 (1.2%) 29 (0.8%)

Penicillin 1,909 (67.1%) 369 (60.8%) 2,278 (66.0%)

Cephems 1,490 (52.4%) 366 (60.3%) 1,856 (53.8%)

Macrolides 668 (23.5%) 145 (23.9%) 813 (23.6%)

Sulfonamides 177 (6.2%) 33 (5.4%) 210 (6.1%)

Quinolones 642 (22.6%) 121 (19.9%) 763 (22.1%)

Others 118 (4.1%) 25 (4.1%) 143 (4.1%)

*: N = number

Because some patients received combination of two or more antibiotics, the sum of each column is over 100%.

Reviewer 2:

I have now concluded the review of your manuscript entitled "Systemic steroid therapy for pneumonic chronic obstructive pulmonary disease exacerbation: A retrospective cohort study."

This study, which investigates the association between systemic steroid therapy and 30-day mortality among patients with pneumonic COPD exacerbations, is undeniably fascinating. The manuscript is generally well-written and provides valuable insights.

However, to enhance the overall quality and impact of your manuscript, I recommend considering the following suggestions:

Response: We thank you for your constructive comments and suggestions for improving our manuscript. We have revised the manuscript accordingly.

1. In the Introduction section, you have referred to several relevant studies. It would be helpful if you could elaborate on the results of these studies, incorporating their effect sizes to provide a comprehensive understanding of the current knowledge in the field.

Response: We have concisely summarized the current evidence in the introduction section so that readers can understand the rationale of our study.

Introduction

Page 4, Lines 69–78: “Several retrospective cohort studies have evaluated the effectiveness of systemic steroid therapy for pneumonic COPD exacerbation [2,7]. A single-center retrospective study by Scholl et al. suggested that the length of hospital stay may be shorter in the non-steroid users (6.0±4.0 days in the steroid group and 4.3±1.8 days in the non-steroid group), whereas there was no difference in 30-day mortality (3/67 [4%] in the steroid group and 0/38 [0%] in the non-steroid group) [7]. The multicenter retrospective study showed inconsistent results: a slightly decreased time to in-hospital death in the main analysis (hazard ratio [steroid/non-steroid group] 0.93, 95% confidence interval [CI], 0.92 to 0.94) and an opposite direction of the treatment effect in a sensitivity analysis (hazard ratio [steroid/non-steroid group] 1.1, 95% CI: 0.53 to 1.5.” 

2. There is a possibility that patients receiving systemic steroids were more severely ill, thereby predisposing them to higher mortality irrespective of steroid therapy. Despite efforts to adjust for known confounders, residual confounding factors might persist. I propose undertaking a sensitivity analysis to account for the severity of illness.

Response: We agree with your comment. Although we used the use of vasopressors and oxygen as proxies for severity on admission, there could be residual confounders. In Japanese administrative claims data, we can collect the ADROP score, a modified version of CURB-65 at admission for patients with a primary diagnosis of bacterial pneumonia. Thus, as a sensitivity analysis, we used the subgroup of patients with the admission-precipitating diagnosis of bacterial pneumonia (ICD-10 codes J12, J13, J14, J15, J16, J18, J69, and P23) comorbid with COPD present at the time of admission (ICD-10 codes J44.1 and J44.9) and a sensitivity analysis additionally incorporating each item of the ADROP score in the main model. The results were similar to those of the main analyses. Finally, we have strengthened the potential residual confounders in the abstract and conclusion section.

Methods

Page 9, Lines 182–190: “In Japanese administrative claims data, we can collect the A-DROP score, a modified version of CURB-65, at admission for patients with primary diagnosis of bacterial pneumonia [20]. Thus, we performed the main analysis among the subgroup of patients with the admission-precipitating diagnosis of bacterial pneumonia (ICD-10 codes J12, J13, J14, J15, J16, J18, J69, and P23) comorbid with COPD present at the time of admission (ICD-10 codes J44.1 and J44.9) and a sensitivity analysis additionally incorporating oxyhemoglobin saturation measured by pulse oximetry ≤90% or partial oxygen pressure in arterial blood ≤60 mmHg and systolic blood pressure ≤90 mmHg extracted from the A-DROP score.”

Discussion

Page 18, Line 315–317: “In addition, we confirmed the same direction of treatment effects among the patients with pneumonia comorbid with COPD and/or COPD exacerbation who had the A-DROP score” 

Abstract

Page 3, Lines 54–55: “Further validation studies based on chart reviews will be needed to cope with residual confounders.”

Conclusion

Page 19, Lines 347–348: “Further validation studies based on clinical records will overcome our study limitation of residual confounders.” 

3. Pneumonic COPD exacerbations are typically attributed to infections. The immunosuppressive effect of systemic steroids could potentially exacerbate these infections. Could you clarify how the potential for increased infection was controlled in your study?

Response: I appreciate your comment. In our study, we could not evaluate the outcome of treatment failure because the database does not have any information on patient improvements in symptoms, vital signs, or imaging results. Thus, we cannot determine why physicians changed the treatment. Although among the patients who were not empirically administered broad-spectrum antibiotics (26.2% of non-systemic steroid users and 20.7% of systemic steroid users), there could be time-varying confounders between the treatment change and the outcomes. We have added this information to the discussion section.

Discussion

Page 17, Lines 292–302: “In addition, among the patients who were not empirically administered broad-spectrum antibiotics, 26.2% of non-systemic steroid users and 20.7% of systemic steroid users were additionally administered broad-spectrum antibiotics two days after admission. We cannot determine whether physicians decided to change the treatment due to an exacerbation of the infection or a subsequent occurrence of a new infection. In addition, there could be time-varying confounders between the treatment change and the outcome. Thus, we could not assess treatment failure as an outcome. Nevertheless, at this point, our results do not support the routine use of systemic steroid therapy on the day of or the day after admission. Evaluation of the effect of systemic steroid therapy during the entire hospital stay requires adjustments for time-varying confounders as well as a much larger sample size.” 

4. In the ‘Statistical Analysis’ section, the paragraph is quite long but there are no references for selecting your method. Please justify yourself about selecting those methods, or refer to some articles on how to select proper regression analysis for continuous independent variables in medical research. Also, reference about propensity score for causal inference and reducing the confounding effects are recommended.

Response: We thank you for pointing this out. We adhered to major statistical guidelines and have added supporting literature in the methods section.

Methods

Pages 8–9, Lines 159–175: “Participants’ characteristics were described by their systemic steroid status as frequencies and proportions for categorical variables and as means with standard deviations or medians and interquartile ranges (IQR) as appropriate for continuous variables. As for the primary outcome, we used a logistic regression model, and for the secondary outcomes, we used a Cox proportional hazard model for time to in-hospital death and a gamma regression model for length of hospital stay [15, 16]. We estimated the average differences in 30-day mortality and length of hospital stay for every single patient based on the presence of systemic steroid therapy. To account for the confounders, we calculated the propensity score, which is the probability of a patient receiving systemic steroid therapy conditioning on confounders, using a logistic regression model with adjustment for known confounders [17]. Each patient was assigned an average treatment-effect weight calculated using the inverse probability of treatment weighting (IPTW) method. We illustrated a love plot to confirm that the absolute standardized mean difference of each covariate was <0.1 in a weighted dataset and a histogram to confirm similar distributions of propensity scores in both treatment groups [18]. We then performed weighted outcome analyses according to the outcome type. Missing data were imputed using multiple imputations by chained equations, which created 100 imputed datasets and combined the estimate within each dataset [19].”

References

15. Faddy M, Nicholas G, Pettitt A. Modeling length of stay in hospital and other right skewed data: comparison of phase-type, gamma and log-normal distributions. Value Health. 2009;12(2):309-314.

16. Lee SW. Kaplan-Meier and Cox proportional hazards regression in survival analysis: statistical standard and guideline of Life Cycle Committee. Life Cycle 2023;3:e8. 

17. Chesnaye NC, Stel VS, Tripepi G, Dekker FW, Fu EL, Zoccali C, et al. An introduction to inverse probability of treatment weighting in observational research. Clin Kidney J. 2021;15(1):14-20. 

18. Stuart EA. Matching methods for causal inference: A review and a look forward. Statist Sci. 2010;25(1): 1-21.

5. Lastly, I would appreciate it if you could highlight the future scope of your study. Your comprehensive limitation discussion indicates several avenues for future research, and these potential directions should be elucidated.

Response: We appreciate your valuable feedback. We have added the subsection to highlight the need for a large-scale prospective study in the future so that researchers can avoid the selection bias of pneumonic COPD and cope with confounding factors more accurately.

Discussion

Pages 18–19 Lines 328–342: “Although a large-scale retrospective study and a randomized controlled study can have additional value, a larger-scale prospective cohort study would be desirable. First, it can avoid misclassification of pneumonic COPD exacerbation. Many physicians do not routinely perform chest computed tomography (CT) scans. In our dataset, pneumonia was detected by chest X-ray in some patients and by chest CT scan in others. The heterogeneity of patients with pneumonic COPD exacerbation would undermine internal and external validity. Researchers should establish a standardized protocol to detect pneumonic COPD exacerbation. In addition, some variables can be collected with difficulty in retrospective studies. For example, no study could use baseline severity of COPD, such as airflow limitation examined by the pulmonary function test and exertional dyspnea examined by the modified Medical Research Council Dyspnea Scale and COPD Assessment Test, because of the large number of missing data. Although a randomized controlled study can overcome the confounding issue by indication, it would require high human and financial costs considering the low effect size of systemic steroid therapy. For these reasons, a large-scale prospective study with a predefined protocol would be the most valuable option.”

---

## [Decision Letter · Decision Letter 1]

13 Aug 2023

Systemic steroid therapy for pneumonic chronic obstructive pulmonary disease exacerbation: A retrospective cohort study

PONE-D-23-10593R1

Dear Dr. Shiroshita,

We’re pleased to inform you that your manuscript has been judged scientifically suitable for publication and will be formally accepted for publication once it meets all outstanding technical requirements.

Kind regards,

Dong Keon Yon, MD, FACAAI, FAAAAI

Academic Editor

PLOS ONE

Additional Editor Comments (optional):

This is an excellent paper.

Reviewers' comments:

Reviewer's Responses to Questions

**Comments to the Author**

1. If the authors have adequately addressed your comments raised in a previous round of review and you feel that this manuscript is now acceptable for publication, you may indicate that here to bypass the “Comments to the Author” section, enter your conflict of interest statement in the “Confidential to Editor” section, and submit your "Accept" recommendation.

Reviewer #1: All comments have been addressed

Reviewer #2: All comments have been addressed

2. Is the manuscript technically sound, and do the data support the conclusions?

Reviewer #1: Yes

Reviewer #2: Yes

3. Has the statistical analysis been performed appropriately and rigorously? 

Reviewer #1: Yes

Reviewer #2: Yes

4. Have the authors made all data underlying the findings in their manuscript fully available?

Reviewer #1: Yes

Reviewer #2: Yes

5. Is the manuscript presented in an intelligible fashion and written in standard English?

Reviewer #1: Yes

Reviewer #2: Yes

6. Review Comments to the Author

Reviewer #1: all comments have been addressed to satisfaction.it is a good article and can be accepted for publication.

Reviewer #2: All comments were addressed. Thank you for authors and editors considering my opinion on this manuscript.

7. PLOS authors have the option to publish the peer review history of their article (what does this mean?). If published, this will include your full peer review and any attached files.

Reviewer #1: No

Reviewer #2: No

---

## [Editor Report · Acceptance letter]

18 Sep 2023

PONE-D-23-10593R1 

Systemic steroid therapy for pneumonic chronic obstructive pulmonary disease exacerbation: A retrospective cohort study 

Dear Dr. Shiroshita:

I'm pleased to inform you that your manuscript has been deemed suitable for publication in PLOS ONE. Congratulations! Your manuscript is now with our production department. 

Kind regards, 

on behalf of

Dr. Dong Keon Yon 

Academic Editor

PLOS ONE